# Machine learning-based prediction of distant metastasis risk in invasive ductal carcinoma of the breast

**Jingru Dong, Ruijiao Lei, Feiyang Ma, Lu Yu, Lanlan Wang, Shangzhi Xu, Yunhua Hu, Jialin Sun, Wenwen Zhang, Haixia Wang, Li Zhang** *

Shihezi University Medical College School of Medical College, Shihezi University, Shihezi, Xinjiang, China

* zl491191385@163.com

**Data Availability Statement:** The original data for this study comes from the SEER database. However, in our research, the data obtained from SEER has been stored in an online at https://

## Abstract

More than 90% of deaths due to breast cancer (BC) are due to metastasis-related complications, with invasive ductal carcinoma (IDC) of the breast being the most common pathologic type of breast cancer and highly susceptible to metastasis to distant organs. BC patients who develop cancer metastases are more likely to have a poor prognosis and poor quality of life, so it is extremely important to recognize and diagnose whether distant metastases have occurred in IDC as early as possible. In this study, we develop a non-invasive breast cancer classification system for detecting cancer metastasis. We used Anaconda-Jupyter notebooks to develop various Python programming modules for text mining, data processing, and machine learning (ML) methods. A risk prediction model was constructed based on four algorithms: Random Forest, XGBoost, Logistic Regression, and SVM. Additionally, we developed a hybrid model based on a voting mechanism using these four algorithms as the base models. The models were compared and evaluated by the following metrics: accuracy, precision, recall, F1-score, and area under the ROC curve (AUC) values. The experimental results show that the hybrid model based on the voting mechanism exhibits the best prediction performance (accuracy: 0.867, precision: 0.929, recall: 0.805, F1-score: 0.856, AUC: 0.94). This stable risk prediction model provides a valuable reference support for doctors in assessing and diagnosing the risk of IDC hematogenous metastasis. It also improves the work efficiency of doctors and strives to provide patients with increased chances of survival.

## 1 Introduction

Breast cancer (BC) is the most common malignant tumor and the main cause of cancer-related deaths in women worldwide [1]. The World Health Organization (WHO) International Agency for Research on Cancer (IARC) released the global cancer burden data in 2020 [2], and the latest statistical report data show that the number of new cases of BC worldwide has exceeded 2.26 million, exceeding the number of cases of lung cancer for the first time. BC has thus been crowned the world's No. 1 cancer, replacing lung cancer, which had been top of the list for many years [2]. In recent years, the incidence and mortality rates of female BC have been increasing yearly, and the proportion of incidence and mortality of all malignant tumors

figshare.com/articles/dataset/_/24996902, allowing readers easier access to the data.

**Funding:** This work was financially supported by the Shihezi University Research Program Project, the fund number is:BJZK202404.

**Competing interests:** The authors have declared that no competing interests exist.

in women has also increased [3], which seriously affects women's physical and mental health. It is projected that by the year 2040, the number of breast cancer patients will increase to 4.07 million, with 1.4 million fatalities [4].

Despite the good therapeutic results and prognosis achieved in the treatment of early-stage BC, the prognosis of patients with metastatic BC is still poor. More than 90% of deaths in patients with BC are not caused by the primary tumor but by distant metastasis of the primary tumor [5, 6]. Infiltrating Ductal Carcinoma (IDC) is the most common subtype among BC, accounting for about 80% of all BC cases [7]. IDC is also the most common form of invasive BC. This pathologic tissue type of BC has a poor prognosis and is highly susceptible to axillary lymph node and distant organ metastases, and the five-year survival rate of patients with distant metastasis from invasive ductal carcinoma is only 27.4%. Therefore, early detection of distant metastasis in BC can help develop individualized treatment plans, which has specific significance for predicting patient survival time and improving patient prognosis [8]. Cancer metastasis risk assessment is important to reduce overall BC mortality. Appropriate risk assessment is a major factor in BC management that determines patient survivability [9].

The successful introduction of information and communication technologies (ICT) in medical practice is an important stake in the renovation of the health system and more precisely in cancer care [10]. With the development of artificial intelligence, machine learning has shown a very strong data mining ability and has been applied to all walks of life. Machine learning is a field that focuses on the learning aspect of artificial intelligence by developing algorithms that best represent a set of data to guide computers to learn from the data and then make the best decisions and predictions based on the results of data analysis [11]. Combining machine learning with medical problems and integrating this development, more intelligent auxiliary diagnosis methods can be proposed to increase treatment time and cure opportunities for patients. In recent years, machine learning has proven to be of great value in BC diagnosis and survival, prognosis prediction, and other aspects [12, 13]. The use of machine learning algorithms to predict the survival risk of BC is currently a research hotspot at the intersection of artificial intelligence and medicine [14].

Although some BC prediction models based on the SEER database have been developed previously (Table 1), few studies have attempted to use the ML algorithm to predict the risk of

**Table 1. Comparison of information on predicting survival outcomes or metastasis of cancer patients using machine learning algorithms in the past.**

| Authors | Prediction model | Objective | Result |
|---|---|---|---|
| Ahmad et al., 2022 [15] | NB, SVM, MLP, J 48 and RF. | A 10-fold cross validation was used. | NB gave the best accuracy. 97.3%. |
| Islam et al., 2020 [16] | SVM, KNN, RF, ANN, and LR | A 10-fold cross validation was employed. | ANN yielded the highest accuracy. 98.57%. |
| Choudhury et al., 2021 [17] | MLP, VP, CC, KLR, SGD, AdaBoost, VFDT, SVM | To diagnose and predict the cancer prognosis of Malignant Pleural Mesothelioma as early as possible (MPM). | AdaBoost have the highest accuracy. 71.29%. |
| Alfian et al., 2022 [18] | SVM | Predicting Breast Cancer from Risk Factors Using SVM and Extra-TreesBased Feature Selection Method. | 80.23% |
| Afolayan et al., 2022 [19] | DT | Breast cancer detection using particle swarm optimization and decision tree machine learning technique. | 92.26% |
| Seo et al., 2022 [20] | SVM | Scaling multi-instance support vector machine to breast cancer detection on the BreaKHis dataset | |
| Ak et al., 2020 [21] | LR | A comparative analysis of breast cancer detection and diagnosis using data visualization and machine learning applications. | 98.1% |
| Sharma and Mishra et al., 2022 [22] | Voting classifier | Performance analysis of machine learning based optimized feature selection approaches for breast cancer diagnosis. | 99.41% |
| Binsaif N et al., 2022 [23] | RF | Application of Machine Learning Models to the Detection of Breast Cancer. | 83.3% |

distant IDC transfers. In this study, machine learning algorithms was used instead of traditional statistical analysis methods to process the clinical characteristics data of patients with IDC, and a machine learning-based predictive model was developed to discover the relationship between certain independent variables and the likelihood of patients with IDC developing distant organ metastasis. Focusing on the predictive models of four algorithms: Logistic Regression, SVM, Random Forest and XGBoost, and using them as base models for integration to construct hybrid models. A ten-fold cross-validation was used to compare the accuracy, recall, precision, and AUC value of each model to predict the risk of cancer metastasis in patients with IDC, thereby assisting clinicians in making more rational clinical decisions and enabling patients to receive treatment earlier, which is useful for breast surgeons to quickly and accurately determine whether cancer metastasis has occurred in patients with IDC.

Our contributions include (i) the use of machine learning algorithms instead of traditional statistical regression methods to process clinical characterization data of patients with IDC to identify patients at high risk of cancer metastasis. (ii) This paper compares several algorithms that can be used to process patient data and determines that a fusion model based on a voting mechanism is the best way to process the data. (iii) It provides a scientific basis and reference for doctors to formulate personalized diagnosis and treatment plans and decisions for patients with different survival prediction results, thus improving treatment outcomes; (iv) It provides a useful reference for the formulation and modification of medical policies.

## 2 System model and methods

### 2.1 Data acquisition

The data used in this experiment were obtained using SEER*stat software (version 8.4.0.1, National Cancer Institute, United States) based on the Surveillance, Epidemiology, and End results (SEER) database, using the "International Classification of Oncological Diseases, Third Edition (ICD-O-3)." The inclusion criteria were (Fig 1): (1) tumor location: site recode ICD-O-3 for breast; (2) pathologic tissue ICD-O-3 staging of 8500/3 (invasive ductal carcinoma); (3) tumor staging using the 8th edition of the American Joint Committee of Cancer (AJCC) TNM staging criteria; (4) availability of complete clinical, pathologic and follow-up data. The exclusion criteria were: (1) missing or unknown disease-related information; (2) age <28 years or >75 years; (3) unknown information such as T, N, and M staging, race, grading, marital status, chemotherapy status, and radiotherapy status; and (4) incomplete follow-up information.

### 2.2 Data pre-processing

Data mining and analysis in this study were done by Python (version 3.10, Python Software Foundation) and SPSS (version 23, IBM, USA). For data preprocessing: (1) Read the dataset: after screening the dataset, there were no missing, abnormal, or duplicated data; (2) Since the positive and negative samples were extremely unbalanced among the included ending variables, this study chose to introduce the function RandomUnderSampler() used to randomly undersample the data from the imblearn library, and resample the data so that the positive to negative sample ratio was 1:1; (3) The categorical feature variables are dummy coded and converted into dummy variables; (4) The data are randomly split into training and test sets in a ratio of 7:3 using train_test_split() in scikit-learn; (5) The transformed dataset is normalized using StandardScaler() to remove the inter-variable scale relationships; (6) Univariate analyses were performed to compare variables between patients who had or had not developed cancer metastases. For categorical data, the chi-square test was used; for continuous non-normally distributed data, the Wilcoxon rank sum test was used. Multivariate logistic regression was

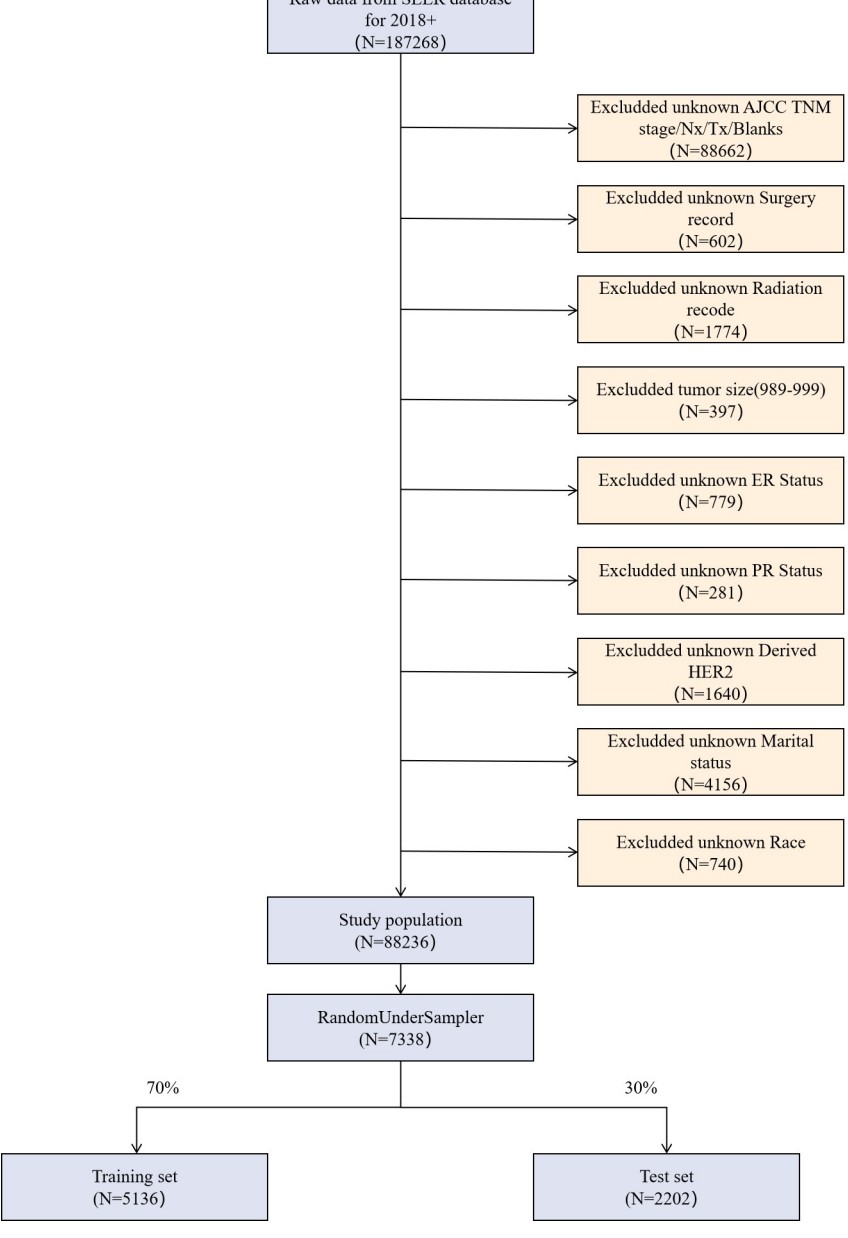

**Fig 1. Data inclusion and exclusion process.**

performed for variables with P < 0.05 in univariate analysis to determine which variables were strongly associated with the development of metastases. Effective data preprocessing is the prerequisite basis for model building and training.

## 2.3 Model establishment

The training and testing procedures of the machine learning models were demonstrated in the JupyterNotebook environment using Python. Python sklearn and pandas software packages were used for data analysis, and Python matplotlib and seaborne were used to draw graphics. The basic modelling process is illustrated in Fig 2. The experimental environment

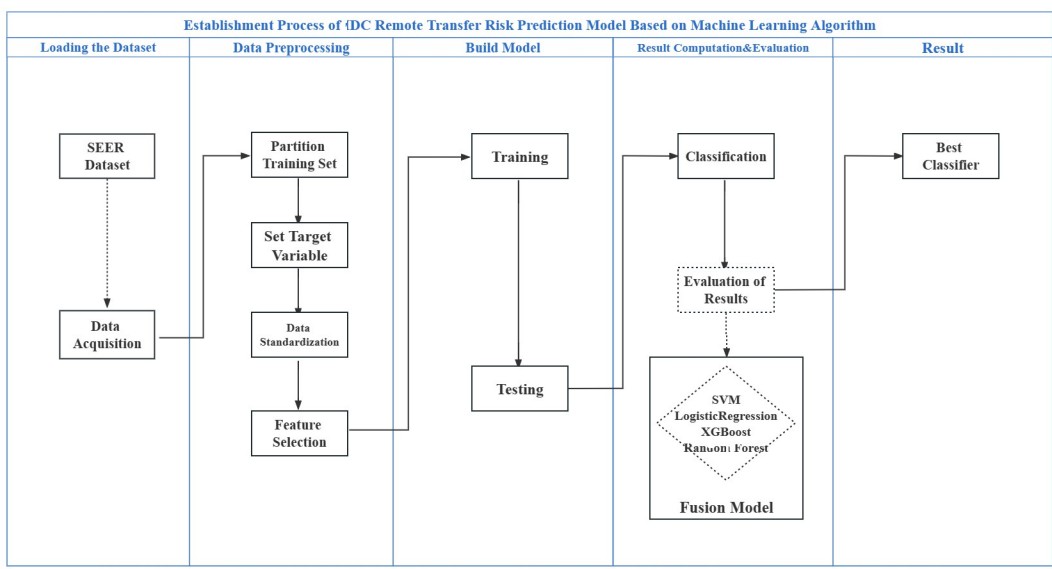

**Fig 2. Modeling flowchart.**

**Table 2. Experimental environment configuration table.**

| Hardware Environment | Software Environment |
|---|---|
| CPU Inter Corel i5–6300hq | Windows 10 |
| GPU GTX960m | Python 3.10 |
| RAM 8GB | PyCharm2021.3.3 |

**Table 3. Detailed information on software packages used for machine learning model development.**

| Package name | Version | Description |
|---|---|---|
| Numpy | 1.25.0 | Numpy is the fundamental package for array computing with python |
| Pandas | 2.1.0 | Powerful data structures for data analysis, time series, and statistics |
| Matplotlib | 3.5.1 | Python plotting package |
| Sklearn | 0.22.0 | A set of python modules for machine learning and data mining |
| Imblearn | 0.11.0 | Toolbox for imbalanced dataset in machine learning |
| Seaborn | 0.12.2 | It provides a high-level interface for drawing attractive and informative statistical graphics. |
| XGBoost | 1.2.0 | XGBoost python package |

configuration and the details of the software packages used for model development are shown in Tables 2 and 3, and the list of abbreviations is shown in Table 4.

Support Vector Machine (SVM) is a supervised learning algorithm whose basic model is a maximally spaced linear classifier defined on the feature space, and whose basic idea is to maximize the marginal distance between the decision hyperplane and the instances closest to the boundary, with the aim of finding the optimal hyperplane that distinguishes between two different classes [24, 25].

**Table 4. List of abbreviations.**

| Full noun name | Abbreviation |
|---|---|
| Accuracy | ACC |
| Area under ROC curve | AUC |
| Logistic Regression | LR |
| Support Vector Machine | SVM |
| EXtreme Gradient Boosting | XGBoost |
| Invasive Ductal Carcinoma | IDC |
| Random Forest | RF |
| Machine Learning | ML |

Logistic Regression (LR) is used to predict the probability of an event occurring by establishing a logistic function (Eq 1) relationship between two or more predictor variables and an outcome variable, i.e., fitting the data to the logistic function. The Sigmoid function introduces a nonlinear element, so it can deal with 0/1 classification problems, and is often used to predict and discriminate the likelihood of a category or event [26, 27].

$$f(z) = \frac{1}{1 + e^{-2}} \tag{1}$$

Random Forest (RF) is a bagging integrated learning algorithm that uses a randomly selected subset of training samples and variables to generate multiple decision trees, each of which is a classifier thatc produces a classification result. By forming a large number of trees into a random forest, the risk of overfitting is reduced and the error converges to some generalized value [4, 28, 29]. The pseudo code of the machine learning model is shown in Table 5, and the basic learner parameters are shown in Table 6.

**Table 5. Pseudocode for machine learning models.**

| Algorithm: Logistic Regression |
|---|
| Step1: For i = 1 to n. |
| Step2: For each data instance, calculate the regression value. |
| Step3: Apply the Sigmoid function to each of obtained regression calculated values. |
| Step4: Finalize the class labels and weights. |
| Step5: Classify, If ($P$>0.5) assign label '1'otherwise, assign label '0'. |

| Algorithm:Linear SVM |
|---|
| Input: Data samples $\{x_i, y_i\}$, i = 1,2,. . .,n, where $x_i \in R^d$, $y_i \in \{-1,+1\}$. |
| Output: Parameters $\omega^*$, $b^*$ and categorical decision function of the categorical hyperplane. |
| Step1: Construct the objective optimization function: |

$$\text{minimize } L = \sum_{ij} \alpha_i \alpha_j y_i y_j x_i^T x_j - \sum_{i=1}^{n} \alpha_i$$

$$\text{s.t} = \sum_{i=1}^{n} \alpha_i y_i = 1$$

$$\alpha_i \geq 0 \ (i = 1,2,. . .,n)$$

| Step2: Find the value of the $\alpha$-vector with the smallest value of the objective optimization function $\alpha^*$ vector; |
|---|
| Step3: Calculate the corresponding coefficients $\omega^*$. |

$$\omega^* = \sum_{i=1}^{n} \alpha^* x_i y_i$$

(*Continued*)

**Table 5.** (Continued)

| Algorithm: Logistic Regression |
|---|

Step4: Find all S sample points that satisfy $\alpha_s > 0$ $\{x_s, y_s\}$, i.e., support vectors, and compute the corresponding parameters $b_s^*$ and all $b_s^*$ of all bs and the average value of all $b_s^*$ $b^*$.

$$b_s^* = y_s - \sum_{n=1}^{n} \alpha_i x_i^T y_i x_s$$

$$b^* = \frac{1}{s} \sum_{i=1}^{s} b_i^*$$

Step5: Output categorical decision function f(x).

$$f(x) = sgn(\omega^* x + b^*)$$

| Algorithm: Random Forest |
|---|

Input: The data sample training set S = $\{x_i, y_i\}$, i = 1,2,..., n, where $x_i \in R^d$, $y_i \in \{-1,+1\}$, and the number of iterations of the base classifier T.

Output: The final strong classifier f(x).

Step1: Perform bootstrap sampling on the original training set S to generate a new training set $S_i$;

Step2: Randomly extract a portion of features from $S_i$ and train the t-th decision tree model. Each tree has no pruning process and grows to the specified tree depth;

Step3: The category with the highest number of votes from the T base classifiers is voted as the final category of the sample to be tested.

EXtreme Gradient Boosting (XGB) is a boosting integrated learning algorithm that is a gradient lifting algorithm based on the CART regression tree. After multiple iterations, each iteration produces a weak classifier, and each classifier is trained based on the gradient of the previous classifier round. The basic idea can be understood as taking the second-order Taylor expansion of the loss function as its replacement function. The objective function of the

**Table 6. Base learner parameters.**

| Model name | parameter |
|---|---|
| Random Forest | N_jobs = 1 |
|  | N_estimators = 500 |
|  | Max_depth = 6 |
|  | Min_samples_leaf = 2 |
|  | Max_features = 'sqrt' |
|  | Verbose = 0 |
| SVM | Kernel = 'linear' |
|  | C = 0.025 |
| XGBoost | N_estimators = 2000 |
|  | Max_depth = 4 |
|  | Min_child_weight = 2 |
|  | Gamma = 0.9subsample = 0.8 |
|  | Objective = 'binary:logistic' |
|  | Scale_pos_weight = 1 |
| Logistic Regression | N_jobs = 1 |
|  | penalty = '12' |
|  | C = 1 |
|  | fit_intercept = True |
|  | intercept_scaling = 1 |
|  | max_iter = 100 |

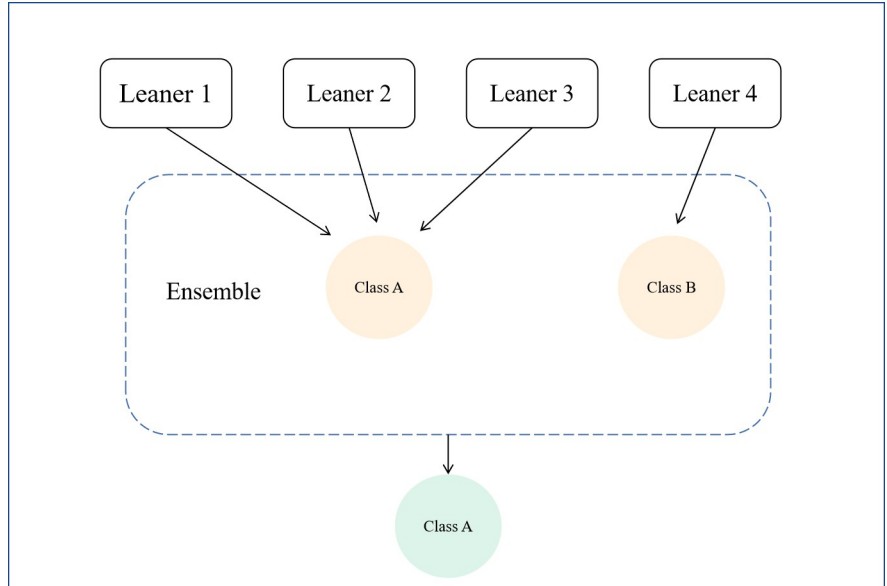

**Fig 3. Schematic diagram of fusion model based on voting mechanism.**

algorithm is composed of the loss function L of the model and the regular term $\Omega$ to restrain the complexity of the model.

The four modeling algorithms mentioned above are built into a fusion model based on the voting mechanism, which is an integrated learning model following the principle of minority-majority, and reduces the variance through the integration of multiple models, thus improving the robustness of the model. Typically in the categorization case, the integrated learner $h_i(x)$ needs to predict a category token from the category labels, and in general, the frequently used method is the voting method (Eq 2). If more than half of the votes are cast for a token, it is expected to be that token; otherwise, the prediction is rejected. Theoretically the final output is the most popular prediction in the base learner. The basic principle is shown in Fig 3.

$$H(x) = \begin{cases} c_i, \text{if } \sum_{i=1}^{T} h_j^i(x) > 0.5 \sum_{k=1}^{N} \sum_{i=1}^{T} h_j^k(x) \\ \text{reject, otherwise} \end{cases} \tag{2}$$

## 2.4 Evaluation indicators

In artificial intelligence, the Confusion Matrix is a common visualization tool in supervised learning. Its main function is to summarize the results of a classifier. The common evaluation criteria used in evaluating the classification performance are: Accuracy, Precision, Recall, F1-score and the ROC curve (Eqs 3–7).

$$\text{Accuracy} = \frac{TP + TN}{TP + FN + FP + TN} \tag{3}$$

$$\text{Precision} = \frac{TP}{TP + FP} \tag{4}$$

$$\text{Recall} = \frac{TP}{TP + FN} \tag{5}$$

$$F1 - score = \frac{2*Precision*Recall}{Precision + Recall} \tag{6}$$

$$AUC = \frac{\sum R_i(I_p) - I_p(I_p + 1)/2}{I_p + I_n} \tag{7}$$

Where TP refers to True Positive, TN refers to True Negative, FP refers to False Positive, and FN refers to False Negative; the Fl-score is a statistical measure of the precision of a dichotomous model, considering both the precision and the recall of the model. A higher F1 score indicates better performance of the model; AUC: The area under the curve is a numerical indicating how well the model will perform in different situations. In general, an AUC greater than 0.85 indicates excellent model performance; Learning Curve is a curve about the sample size of the training datasets, the average score on the training samples and cross-validation samples as well as between the score intervals. By observing the plotted Learning Curve graph, you gain a more intuitive understanding of the state of the model, such as: overfitting or underfitting.

## 3 Result

### 3.1 Demographic characterization

A total of 88,236 patients with confirmed IDC were included in this study from the SEER database, with 14 characteristic variables and outcome variables related to clinicopathology and patient demographics. Demographic variables included race, gender, marital status, and age, while clinicopathological variables included tumor size, estrogen status, progesterone status, HER2 status, TN staging, radiotherapy, and surgical treatment. All data distributions are shown in Table 7. The number of positive and negative samples was equalized by undersampling M1 (3669): M0 (3669) = 1:1. The entire SEER dataset was divided into a training set (n = 5136, 70%) and a test set (n = 2202, 30%). Detailed distribution information is shown in Table 8.

### 3.2 One-way logistic regression analysis

Univariate chi-square test analysis showed (Table 9) that age, gender, race, ER/PR/HER2 status, marital status, tumor size, radiological and surgical treatments, and TN staging in IDC patients were significantly associated with the outcome variable M (P<0.05). Additionally, one-way logistic regression analysis of the significantly different variables showed that HER2 status, tumor size, radiotherapy, chemotherapy, surgical treatment, and TN staging may be risk factors for the development of cancer metastasis in patients with IDC. These six variables were used for further machine learning modeling studies.

### 3.3 Correlation analysis of characteristic variables

The correlation analysis of the feature variables of the datasets demonstrates information about the degree of interaction among these variables. This study uses heat maps to show the correlation of these features with the predictions of the ML algorithm. In the heat map, color indicates the degree of correlation, the darker the color the higher the correlation between the parameters. Fig 4 shows the correlation matrix of 13 variables in the datasets, displaying the correlation between variables. Each cell in the matrix shows the Spearman correlation coefficient between two variables. From the graph, it can be seen that there is a significant negative

**Table 7. Clinical and pathological characteristics of the study subjects.**

| Variables | All | M1 | M0 |
|---|---|---|---|
| | N = 88236 | N = 3669 | N = 84567 |
| **Age** | | | |
| <50 | 19294 (21.9%) | 914 (4.7%) | 18380 (95.3%) |
| >50 | 68942 (78.1%) | 2755 (3.9%) | 66187 (96.0%) |
| **Sex** | | | |
| Female | 87458 (99.1%) | 3621 (4.1%) | 83837 (95.8%) |
| Male | 778 (0.8%) | 48 (6.1%) | 730 (93.8%) |
| **Race** | | | |
| White | 68474 (77.6%) | 2703 (3.9%) | 65771 (96.0%) |
| Black | 9486 (10.7%) | 581 (6.1%) | 8905 (93.8%) |
| American Indian/Alaska Native | 575 (0.6%) | 24 (4.1%) | 551 (95.8%) |
| Asian or Pacific Islander | 9701 (10.9%) | 361 (3.7%) | 9340 (96.2%) |
| **ER Status Recode** | | | |
| Negative | 14662 (16.6%) | 923 (6.2%) | 13739 (93.7%) |
| Positive | 73574 (83.3%) | 2746 (3.7%) | 70828 (96.2%) |
| **PR Status Recode** | | | |
| Negative | 23771 (26.9%) | 1359 (5.8%) | 22376 (94.1%) |
| Positive | 64465 (73.0%) | 2274 (3.5%) | 62191 (96.4%) |
| **HER2 Recode** | | | |
| Negative | 75169 (85.1%) | 2779 (3.6%) | 72390 (96.3%) |
| Positive | 13067 (14.8%) | 890 (6.8%) | 12177 (93.1%) |
| **Marital status** | | | |
| Single | 14303 (16.4%) | 13493 (94.3%) | 810 (5.6%) |
| Married | 51439 (58.3%) | 1768 (3.4%) | 49671 (96.5%) |
| Divorced | 9740 (11.0%) | 461 (4.7%) | 9279 (95.2%) |
| Unmarried or Domestic Partner | 576 (0.7%) | 550 (95.4%) | 26 (4.5%) |
| Widowed | 11203 (12.6%) | 545 (4.8%) | 10658 (95.1%) |
| **Tumor Size(mm)** | | | |
| <100 | 87597 (99.3%) | 3457 (3.9%) | 84140 (96.0%) |
| >100 | 639 (0.7%) | 212 (33.1%) | 427 (66.8%) |
| **Chemotherapy recode** | | | |
| Yes | 34839 (39.4%) | 2722 (7.8%) | 32117 (92.1%) |
| None | 53397 (60.5%) | 947 (1.7%) | 52450 (98.2%) |
| **Radiation recode** | | | |
| Yes | 50357 (57.1%) | 1245 (2.4%) | 49112 (97.5%) |
| None | 37879 (42.9%) | 2424 (6.3%) | 35455 (93.6%) |
| **Surgery recode** | | | |
| Surgery performed | 83233 (94.3%) | 1144 (1.3%) | 82089 (98.6%) |
| Not performed | 5003 (5.6%) | 2525 (50.4%) | 2478 (49.5%) |
| **T stage** | | | |
| T0 | 110 (0.1%) | 30 (27.2%) | 80 (72.7%) |
| T1 | 54376 (61.6%) | 618 (1.1%) | 53758 (98.8%) |
| T2 | 26813 (30.3%) | 1361 (5.0%) | 25452 (94.9%) |
| T3 | 4277 (4.8%) | 648 (5.1%) | 3629 (84.8%) |
| T4 | 2660 (3.0%) | 1012 (38.0%) | 1648 (61.9%) |
| **N stage** | | | |
| N0 | 62875 (71.3%) | 905 (1.4%) | 61970 (98.5%) |

*(Continued)*

**Table 7.** (Continued)

| Variables | All | M1 | M0 |
|---|---|---|---|
| | N = 88236 | N = 3669 | N = 84567 |
| N1 | 20359 (23.1%) | 1688 (9.1%) | 18493 (90.8%) |
| N2 | 2741 (3.1%) | 307 (11.2%) | 2434 (88.7%) |
| N3 | 2261 (2.6%) | 591 (26.1%) | 1670 (73.8%) |

correlation between chemotherapy and TN staging, while T staging is significantly positively correlated with tumor size. In clinical practice, the definition of T staging is based on the diameter of the tumor. This is consistent with clinical experience.

## 3.4 Model performance evaluation

The performance of six machine models was built and compared based on the confusion matrix, learning curves, receiver operating characteristic curves (ROCs), ten-fold cross-validation accuracy, precision, recall, F1-score and AUC value. The results are shown in Figs 5–8. Each model has its relative advantages. Taken together, the fusion model has higher indicators, with accuracy, Recall, Precision, F1-score, and AUC values of 0.867, 0.805, 0.929, 0.856, and 0.94, respectively. One of the biggest drawbacks of the fusion model based on voting mechanism is that if our fusion algorithms are all carefully tuned algorithms, the voting scheme may lead to overfitting. The assessment of the model state was performed using learning curves, and after cross-validation, the fitting effect of RF and SVM is very unsatisfactory. The LR model is the best. Although FM is not the best, it has already reached a relatively ideal state.

## 3.5 Importance analysis of characteristic variables

The PermutationImportance method provided by eli5 in Python is used to calculate the importance of features. Fig 9 shows the feature importance ranking of the five prediction models for cancer metastasis prediction for the 13 feature variables after dimensionality reduction processing on the training set. Variables screened by univariate and multivariate logistic analyses contributed to the predictions in all five models. Surgery ranked first or second in terms of feature importance among all predictive models, indicating that surgery has a strong impact on the prediction of cancer metastasis in IDC. Variables such as age, gender, and marital status always rank in the bottom five, and there is no significant difference in their contribution to the model.

## 4 Discussion

Over the past few decades, the incidence of death in patients with BC has decreased significantly due to advances in surgical methods and systematic neoadjuvant chemotherapy [30]. Metastatic BC, rather than primary tumors, causes more than 90% of cancer-related deaths [31]. As there are no significant symptoms in the early stage of BC, it is easy to be ignored by patients. As a result, the disease has often developed to the middle and late stages when it is found, and recurrence and metastasis are easy to occur after systematic treatment, which is not only difficult to treat, but also greatly affects the quality of life of the patients. In the current treatment process, pathological examination is the gold standard for metastatic tumors. When a physician suspects that a patient has metastatic cancer somewhere, they usually first have the patient undergo a series of immunohistochemical tests to aid in the diagnosis; not only will this invasive test be more physically harmful to the patient, but the pathology results are

**Table 8. Distribution of clinical pathological features in training and testing sets.**

| Variables | Training set (N = 5136) | | Test set (N = 2202) | |
|---|---|---|---|---|
| | M0 (N = 2584) | M1 (N = 2552) | M0 (N = 1085) | M1 (N = 1117) |
| **Age** | | | | |
| <50 | 573 | 643 (25.2%) | 215 (19.8%) | 217 (24.3%) |
| >50 | 2011 | 1909 (74.8%) | 870 (80.2%) | 846 (75.7%) |
| **Sex** | | | | |
| Male | 22 (0.8%) | 31 (1.2%) | 5 (0.4%) | 17 (1.5%) |
| Female | 2562 (99.1%) | 2521 (98.7%) | 1080 (99.5%) | 1100 (98.4%) |
| **Race** | | | | |
| White | 1978 (76.5%) | 410 (16.0%) | 840 (77.4%) | 828 (74.1%) |
| Black | 277 (10.7%) | 1875 (73.4%) | 130 (11.9%) | 171 (15.3%) |
| American Indian/Alaska Native | 24 (0.9%) | 15 (0.6%) | 4 (0.4%) | 9 (10.8%) |
| Asian or Pacific Islander | 305 (11.8%) | 252 (9.8%) | 111 (10.2%) | 109 (9.7%) |
| **ER Status Recode** | | | | |
| Negative | 410 (15.8%) | 655 (25.6%) | 168 (15.4%) | 268 (23.9%) |
| Positive | 2174 (84.1%) | 1897 (74.3%) | 917 (84.5%) | 849 (76.0%) |
| **PR Status Recode** | | | | |
| Negative | 712 (27.5%) | 975 (38.2%) | 288 (26.5%) | 420 (37.6%) |
| Positive | 1872 (72.4%) | 1577 (61.7%) | 797 (73.4%) | 697 (62.3%) |
| **HER2 Recode** | | | | |
| Negative | 2197 (85.0%) | 1918 (75.1%) | 941 (86.7%) | 861 (77.0%) |
| Positive | 387 (14.9%) | 634 (24.8%) | 144 (13.2%) | 256 (22.9%) |
| **Marital status** | | | | |
| Married | 1522 (58.9%) | 1239 (48.6%) | 624 (57.5%) | 529 (47.4%) |
| Single | 400 (15.4%) | 557 (21.8%) | 165 (15.2%) | 253 (22.6%) |
| Divorced | 271 (10.4%) | 325 (12.7%) | 123 (11.3%) | 136 (12.1%) |
| Unmarried or Domestic Partner | 20 (0.8%) | 18 (0.7%) | 5 (0.5%) | 8 (0.7%) |
| Widowed | 331 (12.8%) | 369 (14.4%) | 154 (14.1%) | 176 (15.7%) |
| Separated | 40 (1.5%) | 44 (1.7%) | 14 (1.2%) | 15 (1.3%) |
| **Tumor Size(mm)** | | | | |
| <100 | 2571 (99.5%) | 2403 (94.2%) | 1083 (99.8%) | 1052 (94.2%) |
| >100 | 13 (0.5%) | 149 (5.8%) | 2 (0.2%) | 65 (5.8%) |
| **Chemotherapy recode** | | | | |
| None | 1594 (61.6%) | 656 (25.7%) | 393 (36.2%) | 291 (26.0%) |
| Yes | 990 (38.3%) | 1898 (74.2%) | 692 (63.7%) | 826 (73.9%) |
| **Radiation recode** | | | | |
| None | 1097 (42.5%) | 1702 (66.7%) | 481 (44.3%) | 722 (64.6%) |
| Yes | 1487 (57.5%) | 850 (33.3%) | 604 (55.7%) | 395 (35.4%) |
| **Surgery recode** | | | | |
| None | 79 (3.1%) | 1739 (68.1%) | 41 (3.8%) | 786 (70.4%) |
| Yes | 2505 (96.9%) | 813 (31.8%) | 1044 (96.2%) | 331 (29.6%) |
| **T stage** | | | | |
| T0 | 1 (<0.1%) | 17 (0.6%) | 1 (<0.1%) | 13 (1.1%) |
| T1 | 1628 (63.0%) | 434 (17.0%) | 703 (64.8%) | 184 (16.5%) |
| T2 | 779 (30.1%) | 940 (36.8%) | 313 (28.8%) | 421 (37.6%) |
| T3 | 113 (4.4%) | 451 (17.6%) | 43 (3.9%) | 197 (17.6%) |
| T4 | 63 (2.4%) | 710 (27.8%) | 25 (2.3%) | 302 (27.0%) |
| **N stage** | | | | |

(*Continued*)

**Table 8.** (Continued)

| Variables | Training set (N = 5136) | | Test set (N = 2202) | |
|---|---|---|---|---|
| | M0 (N = 2584) | M1 (N = 2552) | M0 (N = 1085) | M1 (N = 1117) |
| N0 | 1916 (74.1%) | 628 (24.6%) | 813 (74.9%) | 277 (24.8%) |
| N1 | 546 (21.1%) | 1291 (50.6%) | 225 (20.7%) | 575 (51.5%) |
| N2 | 73 (2.8%) | 220 (18.6%) | 24 (2.2%) | 87 (7.8%) |
| N3 | 49 (1.9%) | 413 (16.2%) | 23 (2.1%) | 178 (15.9%) |

usually available 2 to 3 days after the test. This waiting time is critical to improving the survival chances of patients with advanced hematogenous metastatic BC [32]. Therefore, early identification of patients with IDC at risk of developing distant metastases from cancer is extremely important. With the rapid advancements in computer technology and artificial intelligence, it has become a trend to use machine learning to analyze patient data to help doctors make correct judgments [33]. Currently, related scholars have achieved promising results in the prediction of survival rate and survival status of cancer patients using machine learning methods such as random forest, SVM, and neural network [34, 35].

This study uses only clinical and pathological indicators of patients with IDC obtained based on the SEER database to establish a validated prediction model. This model aims to classify and predict whether a patient has cancer metastasis, which can easily predict the risk of organ metastasis without imaging or any laboratory tests. This model can help clinicians make a rapid and reliable diagnosis and improve their work efficiency, and can develop individualized treatment plans for different individuals [36], saving treatment time for patients by identifying the risk of metastasis early, thus reducing mortality.

This study builds a prediction model based on four algorithms: logistic regression, SVM, random forest, and XGBoost. From the results after ten-fold cross-validation, each prediction model has its own advantages and disadvantages. The accuracy and AUC of the XGBoost model predictions are relatively high, which indicates a high degree of overall prediction accuracy, along with the high ability of this classifier to rank samples. However, the precision, recall and F1 value of this model are low, which indicates its low prediction accuracy in positive sample results, and a potential for underdiagnosis. Conversely, the LR and RF models have low metrics and are not suitable as good classifiers for risk prediction. The performance of the fusion model based on the voting mechanism on all the indicators is improved to a certain extent compared to the individual base models. High accuracy indicates that the model can classify patients more accurately, high sensitivity implies a low rate of missed diagnosis, and low error indicates stable predictive performance of the model. In addition, learning curves were used to diagnose the overfitting behavior of the model. As the number of training samples increased, the accuracy gap between RF on the training and cross-validation sets was large, indicating that the models can fit the known data well, but the generalization ability is very poor, which is an overfitting. The learning curves of other models can converge on both the training and validation sets, but SVM is only 0.75, with a slight deviation and unsatisfactory fitting. At this time, FM convergence value was approaching 0.9, indicating that the algorithm itself has good fitting ability, the model's generalization ability is also very strong, and the fitting effect is price ideal. Therefore, the Fusion classifier is more suitable as a good predictive model to predict the risk of IDC distant organ transfer. This study is innovative in using ML algorithms to predict the risk of cancer metastasis in patients with IDC. Currently, a number of academics and researchers in the medical field have used traditional linear statistical models and a single machine learning model to predict the risk of BM in patients with IDC

**Table 9. Univariate and multivariate logistic regression analysis.**

| Variables | Univariate analysis | | Multivariate logistic regression analysis | |
|---|---|---|---|---|
| | Chi-Square | P-value | OR (95%CI) | P-value |
| **Age** | 125.398 | <0.001 | | |
| <50 | | | Reference | |
| >50 | | | 1.031 (0.865–1.229) | 0.743 |
| **Sex** | 5.941 | 0.015 | | |
| Male | | | Reference | |
| Female | | | 1.802 (0.947–3.429) | 0.073 |
| **Race** | 37.240 | <0.001 | | |
| White | | | Reference | |
| Black | | | 1.285 (0.558–2.962) | 0.556 |
| American Indian/Alaska Native | | | 1.153 (0.504–2.639) | 0.736 |
| Asian or Pacific Islander | | | 1.560 (0.695–3.501) | 0.281 |
| **ER Status Recode** | 99.689 | <0.001 | | |
| Negative | | | Reference | |
| Positive | | | 0.943 (0.744–1.195) | 0.627 |
| **PR Status Recode** | 96.711 | <0.001 | | |
| Negative | | | Reference | |
| Positive | | | 1.178 (0.956–1.451) | 0.123 |
| **HER2 Recode** | 112.479 | <0.001 | | |
| Negative | | | Reference | |
| Positive | | | 1.205 (1.000–1.451) | 0.050 |
| **Marital status** | 89.147 | <0.001 | | |
| Married | | | Reference | |
| Single | | | 0.851 (0.672–1.078) | 0.182 |
| Divorced | | | 0.868 (0.454–1.660) | 0.669 |
| Unmarried or Domestic Partner | | | 0.993 (0.755–1.306) | 0.959 |
| Separated | | | 0.899 (0.667–1.212) | 0.484 |
| Widowed | | | 0.910 (0.317–2.087) | 0.825 |
| **Tumor Size(mm)** | 2186.407 | <0.001 | | |
| <100 | | | Reference | |
| >100 | | | 2.513 (1.360–4.644) | 0.003 |
| **Chemotherapy recode** | 991.334 | <0.001 | | |
| Yes | | | Reference | |
| None | | | 2.343 (1.960–2.799) | <0.001 |
| **Radiation recode** | 393.383 | <0.001 | | |
| Yes | | | Reference | |
| None | | | 0.635 (0.546–0.740) | <0.001 |
| **Surgery recode** | 3419.256 | <0.001 | | |
| Yes | | | Reference | |
| None | | | 0.027 (0.022–0.033) | <0.001 |
| **T stage** | 2126.272 | <0.001 | | |
| T0 | | | Reference | |
| T1 | | | 0.155 (0.031–0.785) | 0.897 |
| T2 | | | 0.277 (0.055–1.401) | 0.430 |
| T3 | | | 0.518 (0.101–2.652) | 0.121 |
| T4 | | | 1.115 (0.217–5.736) | 0.024 |
| **N stage** | 1885.641 | <0.001 | | |

*(Continued)*

**Table 9.** (Continued)

| Variables | Univariate analysis | | Multivariate logistic regression analysis | |
|---|---|---|---|---|
| | Chi-Square | P-value | OR (95%CI) | P-value |
| N0 | | | Reference | |
| N1 | | | 2.419 (2.039–2.871) | <0.001 |
| N2 | | | 4.238 (3.105–5.783) | <0.001 |
| N3 | | | 8.785 (6.424–12.015) (6.424–12.015) | <0.001 |

[37–39], but few studies have used multi-model fusion techniques for modeling and prediction. In this study, a fusion machine learning model based on multiple algorithms was developed, which was able to more accurately predict the risk of cancer metastasis in patients with IDC and outperformed other single machine learning models developed in this study.

The ranking importance principle was used to analyze the relative characteristic importance of the variables in each ML model [40]. We found that although the relative feature importance varied slightly across ML models, Surgery, T, N staging, and Tumor Size were the top-ranked variables in most of the models, in contrast to the age, race and ER Status variables, which was the last-ranked variable in most of the models, but also contributed to the models. In this study, five variables, Surgery, T and N staging, Tumor Size, Chemotherapy record, were strong predictors of the risk of distant metastasis of IDC with good predictive value. This suggests

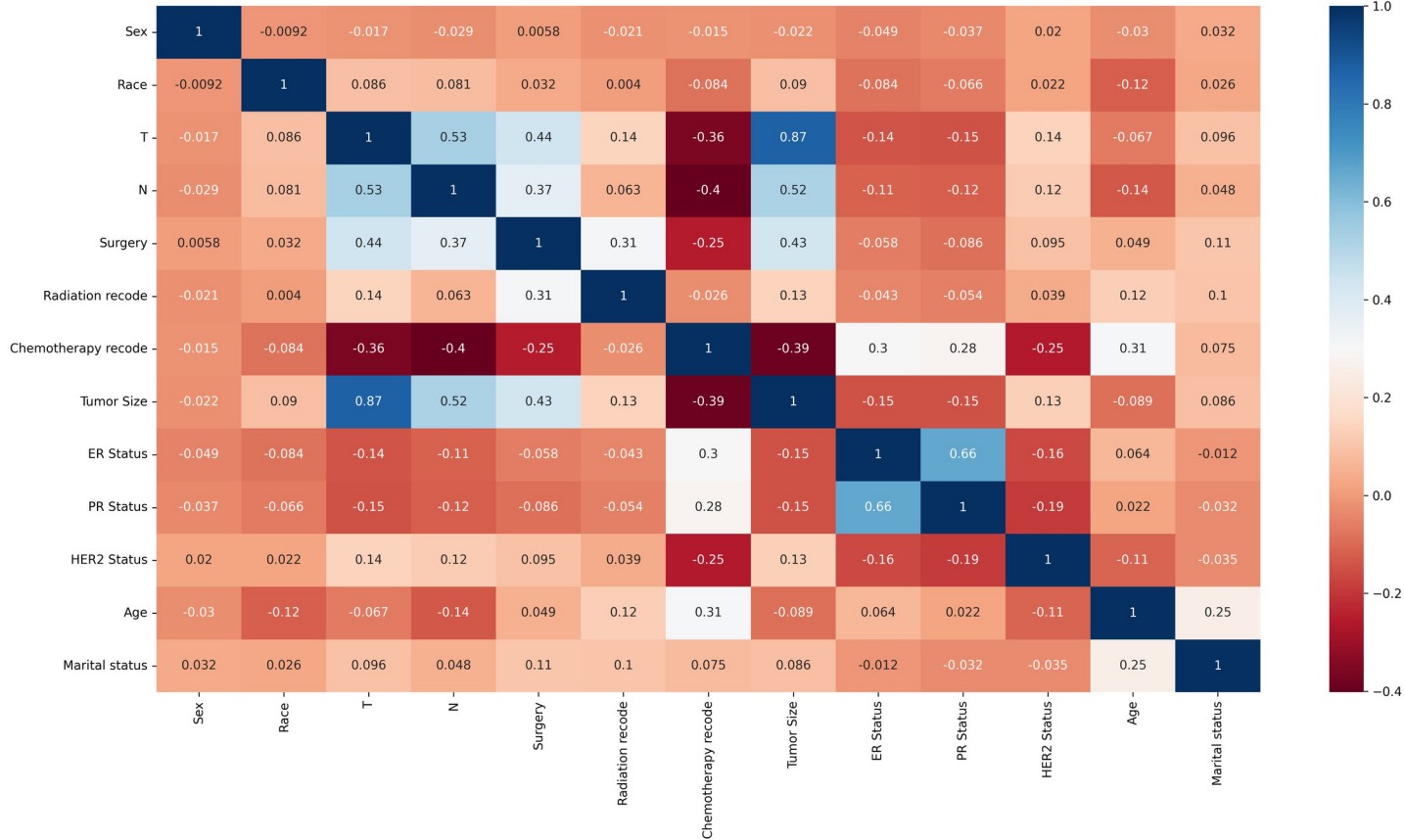

**Fig 4. Heat map of the correlation of features.**

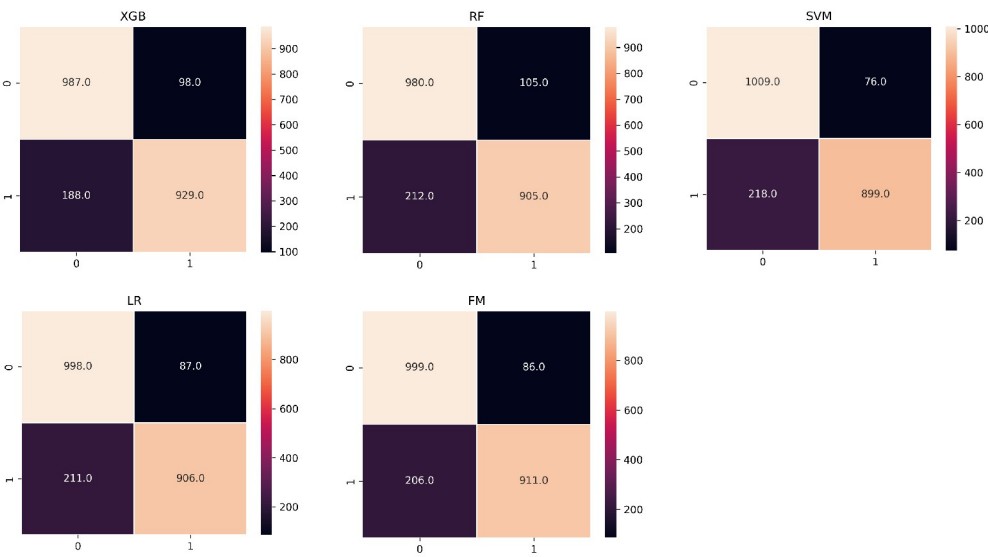

**Fig 5. The confusion matrix of the models in the test set.**

that for which variables were crucial for the creation of the predictive model in this experiment, these characterizing variables could be considered for inclusion in the study when conducting such experiments in the future, to inform the selection of variables in the database.

In the field of medicine, many researchers have attempted to identify independent risk factors that influence the development of cancer metastasis in patients with IDC. In this study, we analyzed the pathological characteristic variables of the patients' based on one-way logistic

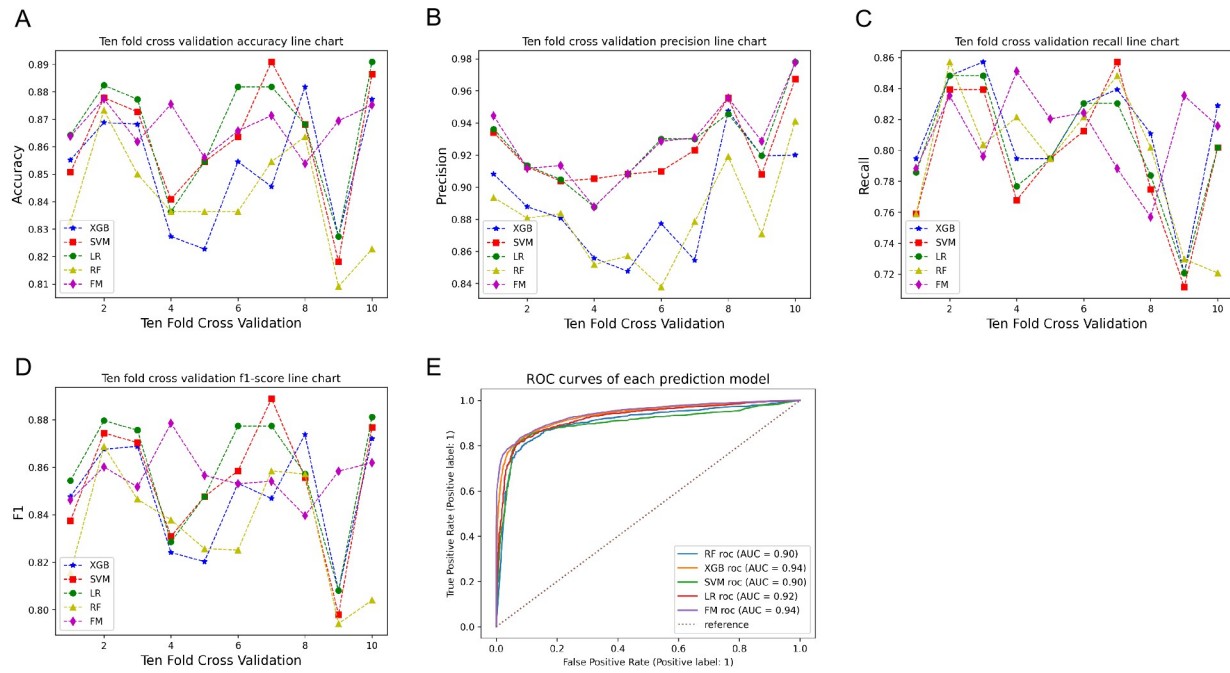

**Fig 6. Comparison of Ten-fold Cross validation metrics for various machine learning models.**

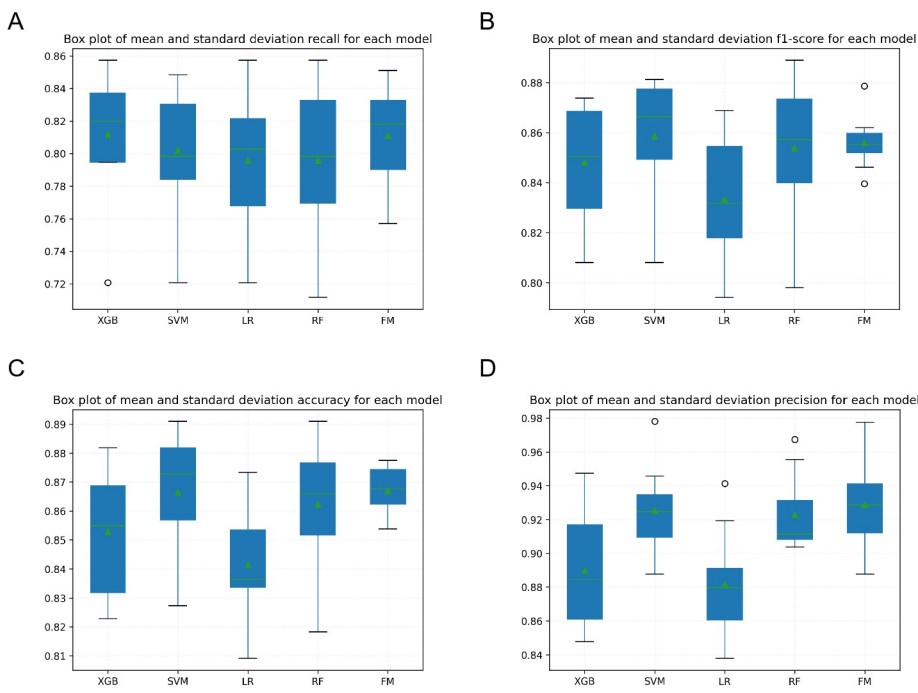

**Fig 7. Box plot comparing the average and standard deviation of indicators.**

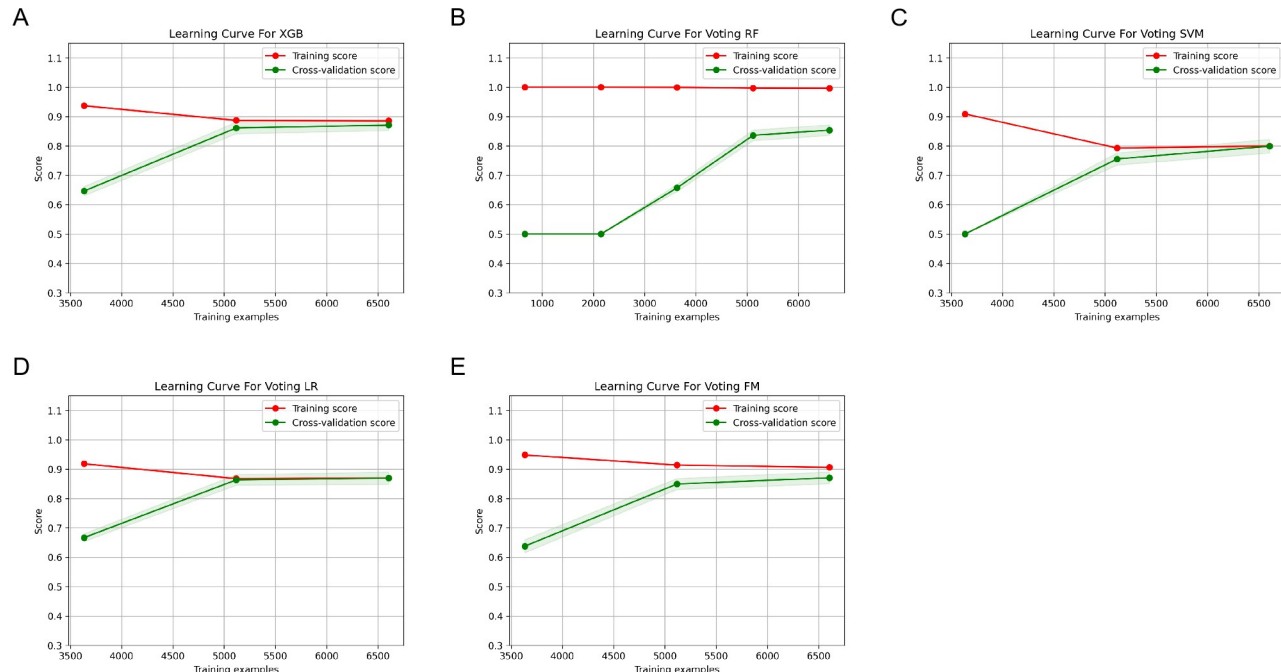

**Fig 8. The learning curve of each machine learning prediction model.**

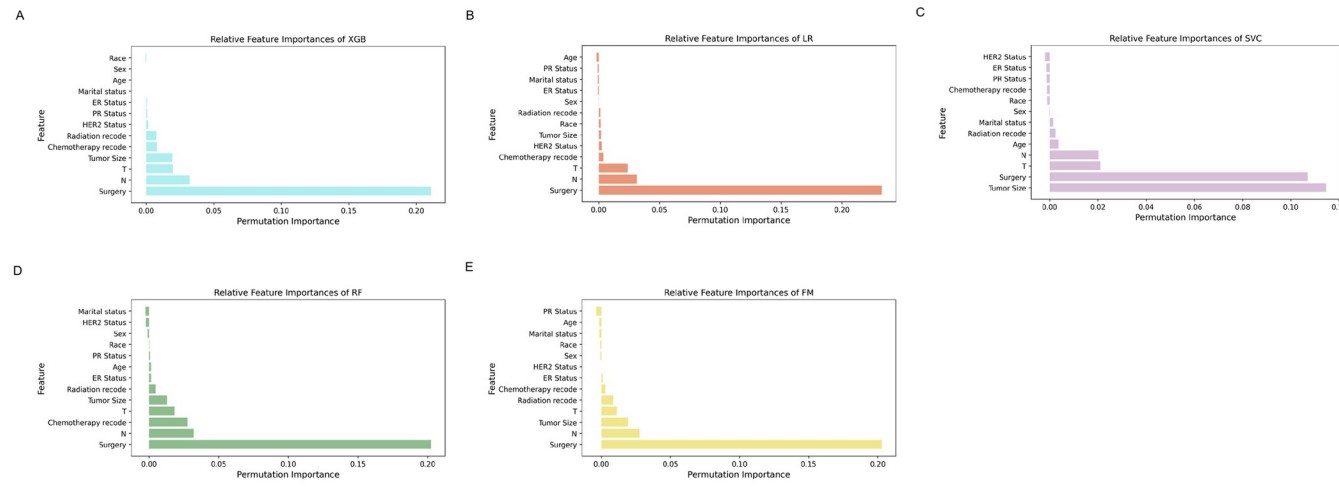

**Fig 9. Contributions of the clinical features variables to risk prediction using feature importance.**

regression and delved into the risk factors affecting the development of cancer metastasis in the patients. We identified that HER2 status, tumor size, radiotherapy, chemotherapy, surgical treatment, and TN staging are risk factors for cancer metastasis in IDC. In clinical practice, patients with low and positive HER2 expression generally have higher tumor grades, larger tumors, more lymph node metastases [41], a smaller percentage of lobular carcinomas [42], and an increased risk of brain metastases [43]. Qi et al. obtained a higher risk of metastasis in HER2-positive patients than in HER2-negative patients in their analysis of risk factors for developing internal breast lymph node metastasis in BC [44], which is consistent with the results of our study. Tumor size was positively correlated with the odds of developing breast tumors. Yazdani et al. found that tumor size was a risk factor for developing bone metastases in BC, and the larger the tumor, the greater the likelihood of bone metastases [45]. In clinical practice, T stage means the size and extent of the primary tumor. In our results, patients with tumor diameters greater than 100 mm and T4 stage were more likely to develop metastasis of cancer cells, which is also consistent with the results of related studies and clinical experience. Poortmans et al [46] reported a lower rate of lymph node metastasis ($<$1.5%) in patients with BC receiving systemic therapy. In our results, patients receiving systemic therapy were less likely to develop cancer metastasis. The results of this study provide a theoretical basis for clinicians to assess the prognosis of patients with IDC, which can be used by physicians to develop more precise and individualized treatment plans for their patients.

However, few studies have focused on combining machine learning to predict the risk of distant metastasis of cancer cells in IDC. Therefore, in this study, we built a model with good predictive performance and good ability to fit the data for predicting the risk of cancer metastasis in IDC patients, based on data from the SEER database and incorporating four machine learning models. Our predictive model allows clinicians to easily input relevant clinical and pathological indicators of their patients into the model, which will calculate the patient's risk of cancer metastasis and then make quick and accurate personalized decisions for their patients based on the predictive output, and can be used as the basis for clinicians to explain their decisions to patients and engage them in their treatment choices. However, our study has certain limitations. Firstly, the data used in this study were limited by the region, quantity, and subjectivity of the feature selection, which limited the final experimental results. Secondly, when artificial intelligence is integrated into the interpretation by clinical physicians, there

may be diverse manifestations in clinical practice [47]. Moreover, this was a retrospective study that needs to be validated through prospective research.

## 5 Conclusion

In this study, we constructed an advanced prediction model that integrates four algorithms: random forest, logistic regression, SVM, and XGB, based on a voting mechanism. This risk prediction model has superior predictive performance and high medical application value. The integrated model based on the voting mechanism shows further improvement upon the already effective predictions of individual models. This enhancement contributes to the improved ability of physicians to accurately identify malignant tumors in breast cancer, holding practical significance for timely and effective treatment for breast cancer patients.

## Supporting information

**S1 Data.**
(ZIP)

## Acknowledgments

The authors would like to thank all the reviewers who participated in the review. We would like to express deep sense of appreciation and heartfelt thanks the public database provided by the National Cancer Institute, American, for providing data support for this study. We would like to thank Editage (www.editage.cn) for English language editing.

## Author Contributions

**Conceptualization:** Haixia Wang.

**Data curation:** Lu Yu.

**Formal analysis:** Lanlan Wang.

**Methodology:** Feiyang Ma.

**Resources:** Li Zhang.

**Software:** Shangzhi Xu.

**Supervision:** Yunhua Hu.

**Validation:** Wenwen Zhang.

**Visualization:** Jialin Sun.

**Writing – original draft:** Jingru Dong.

**Writing – review & editing:** Ruijiao Lei.

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
