## [Decision Letter · Decision Letter 0]

9 Aug 2023

PONE-D-23-20544Machine learning-based prediction of distance metastasis risk in invasive ductal carcinoma of the breastPLOS ONE

Dear Dr. Zhang,

Thank you for submitting your manuscript to PLOS ONE. After careful consideration, we feel that it has merit but does not fully meet PLOS ONE’s publication criteria as it currently stands. Therefore, we invite you to submit a revised version of the manuscript that addresses the points raised during the review process.

ACADEMIC EDITOR: Please revise and resubmit your manuscript.  

We look forward to receiving your revised manuscript.

Kind regards,

Kathiravan Srinivasan

Academic Editor

PLOS ONE

Journal Requirements:

4. Please clarify the Figure 8 "Fig 8. Receiver operating characteristic of each prediction model" in page "15" and Figure 8 "Fig 8. Box plot of mean and standard deviation for each indicator" in page "15".

5. Please upload a copy of Figures 11,12 and 13, to which you refer in your text on page. If 16 and 19the figure is no longer to be included as part of the submission please remove all reference to it within the text.

Reviewers' comments:

Reviewer's Responses to Questions

**Comments to the Author**

1. Is the manuscript technically sound, and do the data support the conclusions?

Reviewer #1: Partly

Reviewer #2: Partly

Reviewer #3: Yes

Reviewer #4: Partly

Reviewer #5: No

Reviewer #6: Partly

2. Has the statistical analysis been performed appropriately and rigorously? 

Reviewer #1: No

Reviewer #2: N/A

Reviewer #3: Yes

Reviewer #4: Yes

Reviewer #5: No

Reviewer #6: I Don't Know

3. Have the authors made all data underlying the findings in their manuscript fully available?

Reviewer #1: Yes

Reviewer #2: Yes

Reviewer #3: Yes

Reviewer #4: Yes

Reviewer #5: No

Reviewer #6: No

4. Is the manuscript presented in an intelligible fashion and written in standard English?

Reviewer #1: No

Reviewer #2: Yes

Reviewer #3: Yes

Reviewer #4: Yes

Reviewer #5: Yes

Reviewer #6: No

5. Review Comments to the Author

Reviewer #1: The study's results were affected by the restricted data availability in terms of regional representation, limited quantity, and subjective feature selection.

-In the context of limited data in this study, data augmentation can be a valuable technique to overcome the scarcity of samples and improve the performance of machine learning models. Data augmentation involves creating additional training examples by applying various transformations to the existing data, while preserving the label information.

For your study, in the case of breast cancer data, data augmentation techniques can be applied to medical images (such as mammograms or MRI scans) to create new images with subtle modifications. These modifications may include rotation, flipping, zooming, translation, or adding noise, among others. By applying such transformations, the dataset can be expanded, enabling the models to learn from a broader range of scenarios.

-In addition to addressing the issue of overfitting, it is essential to thoroughly investigate and fine-tune your model's learning rates.

Reviewer #2: Journal: PLOS ONE

Manuscript Title: Machine learning-based prediction of distance metastasis risk in invasive ductal carcinoma of the breast

Manuscript ID: PONE-D-23-20544

Submission Date: Sunday, July 30, 2023

The manuscript presented “a prediction approach of distance metastasis risk in invasive ductal carcinoma of the breast”. However, the major and critical weak points are that:

(1)Their proposed work discussion is weak distributed to be described or analyzed.

(2)The novelty is not guaranteed.

(3)Their experiments leak from the descriptive and statistical analysis.

The rest of my review presents other weak points, comments, and opinions in detail.

Overall Comments:

(1)[KEYWORDS] The keywords (i.e., index terms) should be sorted in alphabetical order.

(2)[ABSTRACT] The abstract should contain the best-achieved results from the performed experiments.

(3)[ABSTRACT] The abstract should reflect the contributions of the manuscript. I suggest rewriting it.

(4)[INTRODUCTION] The authors should provide a clear problem definition and contributions in the introduction section.

(5)[RESEARCH QUESTION] Where is the research question and research gap?

(6)[RESEARCH QUESTION] The research question is not well-formulated or is poorly motivated, and the paper does not provide new insights or information that is not already known.

(7)[RELATED WORK] Where are the related studies? They should be declared in a separate section.

(8)[RELATED WORK] A table of comparisons should be added at the end of the related studies section to praise the pros. and cons. of them. The year column should be added and they should be ordered by it.

(9)[EQUATIONS] The authors should follow the journal authors’ guidance in writing the equations, symbols, and variables. Please, refer to the authors guidelines on the journal official website.

(10)[EQUATIONS] Where are the equations of the used metrics?

(11)[DATASETS] Samples from the used dataset should be added and annotated.

(12)[METHODOLOGY] The suggested approach is not clearly discussed. More scientific details should be added.

(13)[METHODOLOGY] What are the used equations in the suggested approach? In other words, how the suggested approach is derived?

(14)[METHODOLOGY] Where is the overall pseudocode? Flowchart? of the suggested approach?

(15)[EXPERIMENTS] The working environment (i.e., software and hardware) should be declared and added to a table.

(16)[EXPERIMENTS] The experimental configurations (i.e., settings) should be declared and added to a table.

(17)[EXPERIMENTS] What are the criteria for selecting the experimental configurations?

(18)[EXPERIMENTS] More experiments should be conducted using different configurations.

(19)[EXPERIMENTS] Where is the tabular representation of the reported results?

(20)[EXPERIMENTS] Why did not the authors compare their approach with others in a table?

(21)[EXPERIMENTS] Why did not the authors compare their approach with another approach to compare the suggested approach efficiency and applicability?

(22)[EXPERIMENTS] Where is the detailed and statistical discussion of the reported results?

(23)[EXPERIMENTS] More experiments should be conducted using a different dataset to prove the generalization.

(24)The manuscript does not provide a clear contribution to the field of research.

(25)The manuscript does not provide a clear justification for the research.

(26)[RESULTS] The authors have not provided adequate visual aids, such as graphs or tables, to help readers understand the data.

(27)[ABBREVIATIONS] The authors should add a table of abbreviations in the revised manuscript.

(28)[SYMBOLS] The authors should add a table of symbols in the revised manuscript.

(29)[CONCLUSIONS] The conclusions in this manuscript are primitive. Please, write your conclusions.

(30)[REFERENCES] There are no citations for many sentences in the manuscript. Why? Please check.

(31)[REFERENCES] The references should be written in the same style following the journal authors’ guidance.

(32)[REFERENCES] Recent citations from 2021 and 2022 should be added to the manuscript.

(33)[PROOFING] The authors should get editing help from someone with full professional proficiency in English.

(34)[PROOFING] The manuscript should be checked again to fix any typos such as missing spaces and commas.

(35)[CONSISTENCY] The manuscript structure is too short. It must be elaborated in their applied technology as should support more rigorous technical aspects.

(36)[CONSISTENCY] Some paragraphs are wrapped in more than 10 lines. They should be split concisely.

(37)[NOVELTY] What is the novelty of the suggested approach?

(38)[LIMITATIONS] What are the limitations of the current study? It should be added in a separate section.

For the authors in case of the authors got a chance to review the manuscript and submit the revised one after the editor’s decision, please, provide a table in the revised manuscript mentioning (1) the comment, (2) the authors’ response, and (3) the authors’ change (if applicable). Please, consider all of the comments and don’t ignore any of them.

Please, refer to the attached file "PONE-D-23-20544 Reviewer.pdf" for the same comments in an organized format.

Reviewer #3: Considering the type of study that is retrospective, the project has no ethical problems.

Your project is excellent and I hope that it will increase the life expectancy of cancer patients in the future.

Reviewer #4: First of all, I would like to thanks authors for the efforts they made to make this contribution. The subject is very interesting and has a very relevant impact on society.

Authors try to deal with the metastasis risk in invasive ductal carcinoma of the breast using machine learning techniques,

We have many questions and remarks about the contribution :

- The paper is well-written and easily understandable.

- Authors gave a big part to the Model's definition and presentation, We think they should revisit this part to make it smaller, all the used algorithms are well known.

- A part of the dataset presentation is essential to simplify the problem understanding and make the model's interpretation significant and efficient.

- Are all features important? If Yes why? If No why authors didn't look for a features selection method to optimize the learning process and complexity?

- We think that the authors have to explain more about the added value and the importance of using machine learning in this problem and how their solution can help doctors.

- We recommend that the literature review be in a table containing recent works on the same topics, used models, results, and limitations

- Authors have to present all models configuration, that make the research reproduction possible

- All used features should be clearly presented

- We recommend the use of a schema to present the research process

Wishing all the best for the upcoming research

Reviewer #5: The overall idea of the manuscript to predict the risk of distant organ metastases is good, but the manuscript is not written well and has some flaws:

- The authors did not provide a list of features used to train their model. They need to show the name, type, and specification of features in a table.

- The authors mentioned that they used an 8:2 ratio to split the database into train/test sets, but they did not report any results on the model's performance on the 20% unseen test dataset.

- The authors mentioned in some parts of the manuscript that their original dataset was unbalanced, and they used an undersampling approach to address this issue. However, they did not provide any information on the original size of the dataset or the size of the dataset after downsampling.

- In line 114, the authors mentioned that they used a "small number of features" to train their model. What are the names of these features? They need to be shown in a table.

- The authors claimed that they wanted to predict the "risk" of distant metastases, yet it seems that the models they trained are binary models that have 0/1 output. Usually, for risk prediction, the model needs to have a continuous output to show the probability of an event.

- In Figure 4, what do M0 and M1 represent?

- In Figure 5, the details of the features need to be made clear in a table, as mentioned in the comment above.

- In Figure 6, the confusion matrix needs to be shown in normalized form to make it more readable.

- In Figure 8, a table should be used instead of a figure.

- In line 249, the authors mentioned that they ranked the importance of 14 features. How did they choose these 14 features?

- In line 327, the authors mentioned that there are a few studies that have done the same work. They should reference these studies in their manuscript and compare their approach with theirs.

Reviewer #6: The authors used random forest, XGBoost, logical regression, and support vector machine (SVM) to construct risk prediction models to build a fusion model based on the voting mechanism to develop a prediction model to determine the risk of distant organ metastasis in invasive ductal carcinoma (IDC). Several concerns are raised with this research that should be considered by the authors.

(1)This article's main concern is its innovation. In what way does this research innovate? There have been a lot of studies using the mentioned machine learning models, and the idea of voting is not new. What makes the introduced method superior?

(2)As I read the article, my initial thought was to find a reason, based on the topic's attractiveness, why this article might attract readers. Unfortunately, I didn't find the article to be well organized and I was not justified in reading it.

(3)Despite the fact that a lot of research has been conducted in this field, the authors did not compare other methods. As a result, it cannot be justified.

(4)The abstract of the article has not been prepared properly. The abstract should begin with a short introduction containing the assumption and challenge of the problem, followed by the proposed method to solve the problem, the results (quantitatively and qualitatively), and the conclusion.

(5)There is a lack of clarity and depth in the introduction related work. The authors should organize them professionally. Furthermore, the references are outdated.

(6)Does the authors' assertion that they implemented the method on this dataset justify the method's generalizability?

(7)Results are poorly presented. Can the classification process be affected by changing parameters? Both qualitative and quantitative results are desired by the reader.

(8)The article should be carefully revised by the authors and possible typos and writing errors should be corrected.

(9)In order to understand the disadvantages of the proposed method, it is recommended to include some figures showing that the proposed method does not produce the expected results. It is also better to mention the disadvantages of the proposed method.

(10)The figures (e.g., Figures 8 and 10) are of poor quality. The authors should include more high-quality figures in the text. It is possible to save images in EMF format to ensure better quality.

(11)There will be a better version of this work with better preparation, such as comparative results, different method output figures, etc.

(12)The article appears to have been written in a hurry, has a low level of innovation, and unfortunately, the professor supervisor did not check the article thoroughly. Although this research has some basic flaws, I believe it still faces some basic challenges.

6. PLOS authors have the option to publish the peer review history of their article (what does this mean?). If published, this will include your full peer review and any attached files.

Reviewer #1: No

Reviewer #2: **Yes: **Hossam Balaha

Reviewer #3: No

Reviewer #4: No

Reviewer #5: No

Reviewer #6: No

---

## [Author Response · Author response to Decision Letter 0]

30 Sep 2023

Dear reviewer:

Thank you very much for your comments and professional advice. These opinions help to improve academic rigor of our article. Based on your suggestion and request, we have made corrected modifications on the revised manuscript. Meanwhile, the manuscript had be reviewed and edited by language services of EDITAGE. We hope that our work can be improved again. Furthermore, we would like to show the details as follows:

Reviewer 1#

1.In the context of limited data in this study, data augmentation can be a valuable technique to overcome the scarcity of samples and improve the performance of machine learning models. Data augmentation involves creating additional training examples by applying various transformations to the existing data, while preserving the label information.

For your study, in the case of breast cancer data, data augmentation techniques can be applied to medical images (such as mammograms or MRI scans) to create new images with subtle modifications. These modifications may include rotation, flipping, zooming, translation, or adding noise, among others. By applying such transformations, the dataset can be expanded, enabling the models to learn from a broader range of scenarios.

The author’s answer: Thank you very much for your professional and valuable feedback. We also believe that applying it to images is very meaningful, and we will also consider doing so. However, due to limited research conditions at present. We do not have more relevant data for analysis. In the following research, we will consider expanding the dataset and applying it to a wider range of studies.

2.In addition to addressing the issue of overfitting, it is essential to thoroughly investigate and fine-tune your model's learning rates.

The author’s answer: Thank you very much for your professional advice. In our study, we used different indicators including accuracy, precision, recall, AUC value, F1 value, etc. to evaluate the learning effectiveness of the model. The specific research results are presented in 3.4 Model Performance. In addition, we used undersampling to solve the problem of model overfitting, and then used learning curves to evaluate the final fitting effect.

Reviewer 2#

1.[KEYWORDS] The keywords (i.e., index terms) should be sorted in alphabetical order.

The author’s answer: Thank you for pointing this out. The keywords has been sorted alphabetically and modified.

2.[ABSTRACT] The abstract should contain the best-achieved results from the performed experiments.

3.[ABSTRACT] The abstract should reflect the contributions of the manuscript. I suggest rewriting it.

The author’s answer: Thank you for your suggestion. We have rewritten the abstract section, which includes the best results of the experiments conducted and reflects the contributions of the manuscript. The changes to the manuscript are given in the blue text.

4.[INTRODUCTION] The authors should provide a clear problem definition and contributions in the introduction section.

The author’s answer: Thank you for your professional suggestion. We have made changes to the introduction and provided a definition of the problem, and we provided the contributions of the manuscript at the end of the introduction. The changes and additions to the introduction are given in the yellow text.

5.[RESEARCH QUESTION] Where is the research question and research gap?

6.[RESEARCH QUESTION] The research question is not well-formulated or is poorly motivated, and the paper does not provide new insights or information that is not already known.

The author’s answer: Thank you very much for your professional and valuable feedback. We propose a research gap in the fourth paragraph of the introduction. We have compared (Table 1) with other relevant studies to reflect the current research gap in predicting the risk of IDC cancer metastasis. At the same time, we have proposed our scientific questions to fill this research gap. The changes to the manuscript are given in the yellow text.

7.[RELATED WORK] Where are the related studies? They should be declared in a separate section.

8.[RELATED WORK] A table of comparisons should be added at the end of the related studies section to praise the pros. and cons. of them. The year column should be added and they should be ordered by it.

The author’s answer: Thank you very much for your professional and valuable advice. After reading and consulting many literatures, we sorted out the research of machine learning on breast cancer prediction. We present it in the manuscript through Table 1.

9.[EQUATIONS] The authors should follow the journal authors’ guidance in writing the equations, symbols, and variables. Please, refer to the authors guidelines on the journal official website.

10.[EQUATIONS] Where are the equations of the used metrics?

The author’s answer: Thanks for your careful checks. Based on your comments, we have made the corrections to make the equations conform to the journal author's guide. The equation for the indicators used is given in the section '2.4 Evaluation indicators'.

11.[DATASETS] Samples from the used dataset should be added and annotated.

The author’s answer: Thank you for your comments. We have submitted all datasets in the attachment, including raw data, datasets after inclusion and exclusion criteria, training and testing dataset samples, and undersampling dataset samples.

12.[METHODOLOGY] The suggested approach is not clearly discussed. More scientific details should be added.

13.[METHODOLOGY] What are the used equations in the suggested approach? In other words, how the suggested approach is derived?

14.[METHODOLOGY] Where is the overall pseudocode? Flowchart? of the suggested approach?

The author’s answer: We sincerely appreciate the valuable comments. We discussed the proposed method in detail in '2.3 Model Establishment', adding some scientific details and main equations (Equation 2). The overall pseudo code is presented in Table 5.

15.[EXPERIMENTS] The working environment (i.e., software and hardware) should be declared and added to a table.

16.[EXPERIMENTS] The experimental configurations (i.e., settings) should be declared and added to a table.

17.[EXPERIMENTS] What are the criteria for selecting the experimental configurations?

18.[EXPERIMENTS] More experiments should be conducted using different configurations.

19.[EXPERIMENTS] Where is the tabular representation of the reported results?

20.[EXPERIMENTS] Why did not the authors compare their approach with others in a table?

21.[EXPERIMENTS] Why did not the authors compare their approach with another approach to compare the suggested approach efficiency and applicability?

22.[EXPERIMENTS] Where is the detailed and statistical discussion of the reported results?

23.[EXPERIMENTS] More experiments should be conducted using a different dataset to prove the generalization.

The author’s answer: We sincerely appreciate the valuable comments. For the 'EXPERIMENTS' question, we provide the following answers: 

[15-16] The experimental environment configuration, detailed information on the software package used for machine learning model development, and basic learner parameters information are added to Tables 3, 4, and 6, respectively.

[17] The criteria for selecting experimental configurations are to strive to achieve optimal model performance by setting the above parameters during model training.

[18] We used different parameters to conduct more experiments during the code running process, and ultimately presented the best results in the article.

[19] Thank you very much for your proposal. We have added Tables 7, 8, and 9 to the results section to illustrate the distribution information of the data. Figures 5-8 are the main results of evaluating and comparing the performance of the model.

[20-21] Thank you very much for your professional and valuable advice. After reading and consulting many literatures, we sorted out the research of machine learning on breast cancer prediction. We present it in the manuscript through Table 1.

[22] Thank you for your comments. We conducted a more detailed and in-depth discussion on the results of the report in the discussion section. We updated the background description of breast cancer with reference to the latest literature (273-278 lines); We reorganize and analyze the results (lines 300-322); Discuss in detail the results of univariate and multivariate logistic regression analysis (lines 343-367). The additions to the discussion are given in the red text, and the changes to the discussion are given in the green text.

[23] We appreciate your careful reading of our manuscript and your professional feedback. We tested the model in our study using data from only 20% of the internal test set. There is no external dataset to validate it, which is the main limitation of this study. We strongly agree to use some external data to validate our model. We apologize for the fact that there is currently no available data to conduct the experiments. We will definitely address this limitation in the future if we have the opportunity.

24.The manuscript does not provide a clear contribution to the field of research.

The author’s answer: We explained in the last paragraph of the introduction the contribution of the manuscript to the research field. The additions to the discussion are given in the blue text(lines 292-299).

25.The manuscript does not provide a clear justification for the research.

The author’s answer: Thank you very much for your question. Our study aims to establish an efficient machine learning model to predict the risk of cancer metastasis in IDC patients. Thereby assisting clinicians in making more rational clinical decisions and enabling patients to receive treatment earlier, which is useful for breast surgeons to quickly and accurately determine whether cancer metastasis has occurred in patients with IDC. We have introduced both the introduction and discussion sections. In addition, we also focused on the performance of single machine learning models and multi algorithm fusion machine learning models, and found that the fusion model has stronger learning performance.

[RESULTS] The authors have not provided adequate visual aids, such as graphs or tables, to help readers understand the data.

The author’s answer: Thank you very much for your comments. Based on your suggestion, we have added more tables in the article, such as Tables 7-9. Compared to the initial draft, we have added a total of 9 tables as visual aids, which can help readers understand the data.

26.[ABBREVIATIONS] The authors should add a table of abbreviations in the revised manuscript.

27.[SYMBOLS] The authors should add a table of symbols in the revised manuscript.

The author’s answer: Thank you for your suggestion. We have added a list of abbreviations (Table 2).

28.[CONCLUSIONS] The conclusions in this manuscript are primitive. Please, write your conclusions.

The author’s answer: Thank you very much for your comments and professional advice. We have made changes to the conclusion section to make it more concise and highlight the key points.

29.[REFERENCES] There are no citations for many sentences in the manuscript. Why? Please check.

30.[REFERENCES] The references should be written in the same style following the journal authors’ guidance.

31.[REFERENCES] Recent citations from 2021 and 2022 should be added to the manuscript.

The author’s answer: We are really sorry for our careless mistakes. We have re reviewed the article and corrected this error. Added citation identification to quoted sentences. And modify the reference format according to the requirements of the journal author's guide. Added some latest references from 2021 and 2022. Such as references 1, 2, 13, 18, 23, 30, 33, 35, 36, etc.

32.[PROOFING] The authors should get editing help from someone with full professional proficiency in English.

33.[PROOFING] The manuscript should be checked again to fix any typos such as missing spaces and commas.

The author’s answer: Thanks for your careful checks. We are sorry for our carelessness. We first rechecked the manuscript and corrected typos, punctuation errors, and missing spaces. We have invited a professional polishing agency to help polish our manuscript. The editing certification will be provided in the attachment. And we hope the revised manuscript could be acceptable for you.

34.[CONSISTENCY] The manuscript structure is too short. It must be elaborated in their applied technology as should support more rigorous technical aspects.

35.[CONSISTENCY] Some paragraphs are wrapped in more than 10 lines. They should be split concisely.

The author’s answer: We greatly appreciate your valuable advice. Based on your suggestion, we have added some article structures. For example, in the results section, we have added '3.1 Demographic characterization' and '3.2 One-way logistic regression analysis' to make the manuscript richer in content and more complete in structure. We have re edited some paragraphs in the manuscript with more than 10 lines to make them concise and concise.

36.[NOVELTY] What is the novelty of the suggested approach?

The author’s answer: As each learning model has its own advantages and disadvantages. For example, the XGBoost algorithm adds a regular term to the objective function to control the complexity of the model. It makes the learned model simpler and prevents overfitting. However, this modeling algorithm has too many parameters and is complex to tune. On the contrary, random forest learning model training can be highly parallelized, fast training speed, high efficiency. However, it is prone to overfitting in some noisy classification problems and or regression problems. Then, by constructing a machine learning model that fuses several models together, we can maximize the guarantee that this new model has the maximum advantages and minimum disadvantages. In addition, to the best of our knowledge, it is relatively rare to use this model for predicting IDC transfer risk.

37.[LIMITATIONS] What are the limitations of the current study? It should be added in a separate section.

The author’s answer: The main limitation of this study is that the data used are limited by region, volume and subjectivity of feature selection, which constrains the final experimental results. In addition, this is a retrospective study that needs to be validated by prospective studies. We have placed research limitations in a separate section, see '6 Research limitations' in the manuscript.

Reviewer 3#

Considering the type of study that is retrospective, the project has no ethical problems. Your project is excellent and I hope that it will increase the life expectancy of cancer patients in the future.

The author’s answer: Thank you very much for your endorsement of our project. We are very honored by your praise. We have reviewed this article again and carefully revised and supplemented it, making every effort to make it more scientific and rigorous.

Reviewer 4#

1.The paper is well-written and easily understandable.

The author’s answer: We feel very honored by your compliments. We have re-examined the article and made careful changes and additions. We'll keep working on it.

2.Authors gave a big part to the Model's definition and presentation, We think they should revisit this part to make it smaller, all the used algorithms are well known.

The author’s answer: We think this is an excellent suggestion. We have made changes to the presentation part of the model's definition. In the part of the manuscript where the font is marked blue.

3.A part of the dataset presentation is essential to simplify the problem understanding and make the model's interpretation significant and efficient.

The author’s answer: Thanks for your suggestion. As per your suggestion, we have shown the information about the dataset in the section '3.1 Demographic characterization' through Table 6 and Table 7. In addition, the original dataset information, training set and test set data are uploaded through attachments.

4.Are all features important? If Yes why? If No why authors didn't look for a features selection method to optimize the learning process and complexity?

The author’s answer: Thank you very much for your question. We explain the issue as follows: 

---

## [Decision Letter · Decision Letter 1]

24 Oct 2023

PONE-D-23-20544R1Machine learning-based prediction of distant metastasis risk in invasive ductal carcinoma of the breastPLOS ONE

Dear Dr. Zhang,

Thank you for submitting your manuscript to PLOS ONE. After careful consideration, we feel that it has merit but does not fully meet PLOS ONE’s publication criteria as it currently stands. Therefore, we invite you to submit a revised version of the manuscript that addresses the points raised during the review process.

ACADEMIC EDITOR: Please revise and resubmit your manuscript.. 

We look forward to receiving your revised manuscript.

Kind regards,

Kathiravan Srinivasan

Academic Editor

PLOS ONE

Journal Requirements:

Reviewers' comments:

Reviewer's Responses to Questions

**Comments to the Author**

1. If the authors have adequately addressed your comments raised in a previous round of review and you feel that this manuscript is now acceptable for publication, you may indicate that here to bypass the “Comments to the Author” section, enter your conflict of interest statement in the “Confidential to Editor” section, and submit your "Accept" recommendation.

Reviewer #2: All comments have been addressed

Reviewer #3: All comments have been addressed

Reviewer #5: All comments have been addressed

Reviewer #6: All comments have been addressed

2. Is the manuscript technically sound, and do the data support the conclusions?

Reviewer #2: Yes

Reviewer #3: Yes

Reviewer #5: Partly

Reviewer #6: Yes

3. Has the statistical analysis been performed appropriately and rigorously? 

Reviewer #2: No

Reviewer #3: I Don't Know

Reviewer #5: Yes

Reviewer #6: I Don't Know

4. Have the authors made all data underlying the findings in their manuscript fully available?

Reviewer #2: Yes

Reviewer #3: Yes

Reviewer #5: No

Reviewer #6: No

5. Is the manuscript presented in an intelligible fashion and written in standard English?

Reviewer #2: Yes

Reviewer #3: Yes

Reviewer #5: No

Reviewer #6: Yes

6. Review Comments to the Author

Reviewer #2: Journal: PLOS ONE

Manuscript Title: Machine learning-based prediction of distant metastasis risk in invasive ductal carcinoma of the breast

Manuscript ID: PONE-D-23-20544R1

Submission Date: Monday, October 9, 2023

The authors have made a suitable major revision. However, there are some concerns that should be addressed:

Overall Comments:

(1)The conclusions need to be enhanced and the future work should be addressed.

(2)The statistical analysis should be more descriptive.

(3)The following references can be addressed in your study:

a.Balaha, H. M., Antar, E. R., Saafan, M. M., & El-Gendy, E. M. (2023). A comprehensive framework towards segmenting and classifying breast cancer patients using deep learning and Aquila optimizer. Journal of Ambient Intelligence and Humanized Computing, 14(6), 7897-7917.

b.Lee, S. E., Yoon, J. H., Son, N. H., Han, K., & Moon, H. J. (2023). Screening in Patients With Dense Breasts: Comparison of Mammography, Artificial Intelligence, and Supplementary Ultrasound. American Journal of Roentgenology.

c.Baghdadi, N. A., Malki, A., Balaha, H. M., AbdulAzeem, Y., Badawy, M., & Elhosseini, M. (2022). Classification of breast cancer using a manta-ray foraging optimized transfer learning framework. PeerJ Computer Science, 8, e1054.

For the authors in case of the authors got a chance to review the manuscript and submit the revised one after the editor’s decision, please, provide a table in the revised manuscript mentioning (1) the comment, (2) the authors’ response, and (3) the authors’ change (if applicable). Please, consider all of the comments and don’t ignore any of them.

Please, refer to the attached file "PONE-D-23-20544R1 Reviewer.pdf" for the same comments in an organized format.

Reviewer #3: According to the corrections that were suggested by different reviewers , this article seems to be suitable for publishing.

Reviewer #5: Thanks. I didn't see the "data availability" exploitation in the manuscript. That would be good to add a statement in the manuscript to show how readers can access to the data.

Reviewer #6: There have been many solutions to the previous problems. The authors should take care that the responsibility for the results and validation of the results lies with them alone. Figures and references should be carefully reviewed. The references to the equations, as well as the definitions of some of their parameters, should also be carefully checked.

7. PLOS authors have the option to publish the peer review history of their article (what does this mean?). If published, this will include your full peer review and any attached files.

Reviewer #2: **Yes: **Hossam Magdy Balaha

Reviewer #3: No

Reviewer #5: No

Reviewer #6: No

---

## [Author Response · Author response to Decision Letter 1]

15 Jan 2024

Dear reviewer:

Thank you very much for your comments and professional advice. These opinions help to improve academic rigor of our article. Based on your suggestion and request, we have made corrected modifications on the revised manuscript. We hope that our work can be improved again. Furthermore, we would like to show the details as follows:

Reviewer #2

1.The conclusions need to be enhanced and the future work should be addressed.

The author’s answer: Thank you very much for your professional and valuable feedback. We have rewritten the conclusion section to make it more comprehensive. The changes to the manuscript are given in the blue text. As for the next steps, our primary focus will be on external validation, a task we are actively pursuing.

2.The statistical analysis should be more descriptive.

The author’s answer: Thank you very much for the positive and constructive comments regarding our paper. We provided a detailed description of the methodology section, covering steps such as data acquisition, data inclusion, exclusion, and preprocessing, as well as model establishment. To ensure readers have a clear understanding of our statistical analysis workflow, we utilized figures for a more concise presentation of results, such as Figures 1, 2, and 3.

3.The following references can be addressed in your study:

a.Balaha, H. M., Antar, E. R., Saafan, M. M., & El-Gendy, E. M. (2023). A comprehensive framework towards segmenting and classifying breast cancer patients using deep learning and Aquila optimizer. Journal of Ambient Intelligence and Humanized Computing, 14(6), 7897-7917.

b.Lee, S. E., Yoon, J. H., Son, N. H., Han, K., & Moon, H. J. (2023). Screening in Patients With Dense Breasts: Comparison of Mammography, Artificial Intelligence, and Supplementary Ultrasound. American Journal of Roentgenology.

c.Baghdadi, N. A., Malki, A., Balaha, H. M., AbdulAzeem, Y., Badawy, M., & Elhosseini, M. (2022). Classification of breast cancer using a manta-ray foraging optimized transfer learning framework. PeerJ Computer Science, 8, e1054.

The author’s answer: We sincerely appreciate the valuable comments. As suggested by the reviewer, we consider the references you recommended to be valuable for our research. Therefore, we have cited them in the paper (references 2, 5, and 49, respectively).

Reviewer #3

According to the corrections that were suggested by different reviewers , this article seems to be suitable for publishing.

The author’s answer: We would like to thank you for your professional review work, and for the positive feedback.

Reviewer #5

Thanks. I didn't see the "data availability" exploitation in the manuscript. That would be good to add a statement in the manuscript to show how readers can access to the data.

The author’s answer: We thank the reviewer for bringing this to our attention. We have appended a "data availability" statement at the end of the conclusion section and uploaded the data to Figshare. The original data can be directly accessed through the link we provided.

Reviewer #6

There have been many solutions to the previous problems. The authors should take care that the responsibility for the results and validation of the results lies with them alone. Figures and references should be carefully reviewed. The references to the equations, as well as the definitions of some of their parameters, should also be carefully checked.

The author’s answer: We thank the reviewer for the helpful comments and appreciation of our work. We meticulously reviewed the figures, reference materials, as well as the references for equations and definitions of certain parameters, making appropriate modifications. Thanks to the reviewer for this careful comments.

---

## [Decision Letter · Decision Letter 2]

2 Sep 2024

Machine learning-based prediction of distant metastasis risk in invasive ductal carcinoma of the breast

PONE-D-23-20544R2

Dear Dr. Zhang,

We’re pleased to inform you that your manuscript has been judged scientifically suitable for publication and will be formally accepted for publication once it meets all outstanding technical requirements.

Kind regards,

Reza Rabiei

Academic Editor

PLOS ONE

Additional Editor Comments (optional):

Reviewers' comments:

Reviewer's Responses to Questions

**Comments to the Author**

1. If the authors have adequately addressed your comments raised in a previous round of review and you feel that this manuscript is now acceptable for publication, you may indicate that here to bypass the “Comments to the Author” section, enter your conflict of interest statement in the “Confidential to Editor” section, and submit your "Accept" recommendation.

Reviewer #5: All comments have been addressed

Reviewer #6: All comments have been addressed

2. Is the manuscript technically sound, and do the data support the conclusions?

Reviewer #5: Yes

Reviewer #6: Yes

3. Has the statistical analysis been performed appropriately and rigorously? 

Reviewer #5: Yes

Reviewer #6: I Don't Know

4. Have the authors made all data underlying the findings in their manuscript fully available?

Reviewer #5: Yes

Reviewer #6: Yes

5. Is the manuscript presented in an intelligible fashion and written in standard English?

Reviewer #5: Yes

Reviewer #6: Yes

6. Review Comments to the Author

Reviewer #5: Thanks for fulfilling the comments. I believe the manuscript is ready to publish in the current format.

Reviewer #6: As a result of the different opinions of the referees and their comments, the article has been significantly improved and is suitable for publication.

7. PLOS authors have the option to publish the peer review history of their article (what does this mean?). If published, this will include your full peer review and any attached files.

Reviewer #5: No

Reviewer #6: No

---

## [Editor Report · Acceptance letter]

12 Sep 2024

PONE-D-23-20544R2 

PLOS ONE

Dear Dr. Zhang, 

I'm pleased to inform you that your manuscript has been deemed suitable for publication in PLOS ONE. Congratulations! Your manuscript is now being handed over to our production team.

Kind regards, 

on behalf of

Dr. Reza Rabiei 

Academic Editor

PLOS ONE